# Effects of Total Flavonoids of *Artemisia ordosica* on Growth Performance, Oxidative Stress, and Antioxidant Status of Lipopolysaccharide-Challenged Broilers

**DOI:** 10.3390/antiox11101985

**Published:** 2022-10-05

**Authors:** Lulu Shi, Xiao Jin, Yuanqing Xu, Yuanyuan Xing, Sumei Yan, Yanfei Guo, Yuchen Cheng, Binlin Shi

**Affiliations:** College of Animal Science, Inner Mongolia Agricultural University, Hohhot 010018, China

**Keywords:** total flavonoids of *Artemisia ordosica*, broiler, lipopolysaccharide challenge, growth performance, antioxidant capacity

## Abstract

*Artemisia ordosica* has been applied as a traditional Chinese/Mongolian medicine for the treatment of certain inflammatory ailments. This study was conducted to investigate the effect of *Artemisia ordosica* total flavonoids (ATF) supplemented in diets on growth performance, oxidative stress, and antioxidant status in lipopolysaccharide (LPS)-challenged broilers. A total of 240 one-day-old Arbor Acre broilers were randomly allotted into 5 groups with 6 replicates (*n* = 8), which were the basal diet group (CON), LPS-challenged and basal diet group (LPS), and the LPS-challenged and basal diet added with low (500 mg/kg), middle (750 mg/kg), and high (1000 mg/kg) doses of ATF groups (ATF-L, ATF-M, and ATF-H), respectively. On day 16, 18, 20, 22, 24, 26, and 28, broilers were injected intra-abdominally either with LPS or an equivalent amount of saline. Results showed that dietary ATF alleviated the LPS-induced decrease in BW, ADG, and ADFI in broilers. Dietary ATF supplementation reversed the increased serum oxidative damage indexes (reactive oxygen species, protein carbonyl, and 8-hydroxy-2-deoxyguanosine) and the decreased serum antioxidant indexes [total superoxide dismutase (SOD), catalase (CAT), glutathione peroxidase (GPx), and total antioxidant capacity (TAC)] in LPS-challenged broilers. Moreover, ATF alleviated the decreased antioxidase activity and the over-production of malondialdehyde (MDA) in the liver and spleen induced by LPS. This study also showed that ATF alleviated the increased mRNA expression of Kelch-like ECH-associated protein 1 (*Keap1*) and the decreased mRNA expression of nuclear factor erythroid 2-related factor 2 (*Nrf2*), *CAT*, *SOD*, and *GPx* in the liver and spleen of broilers challenged with LPS. In conclusion, ATF has a strong capacity to enhance antioxidant enzyme activity and relieve oxidative stress and can be used as a potential novel feed additive in poultry diets to improve growth performance and antioxidant capacity.

## 1. Introduction

Over the past decades, the modern broiler farming business has grown rapidly because of the advantages of poultry meat, including high production efficiency, low cost, lack of religious restriction, and multiple consumption options [1,2]. Referring to related data and reports, the global demand for poultry meat is likely to grow at a peak rate of 121% between 2005 and 2050 [3]. Admittedly, broilers can reach market weight in a shorter period of time due to genetic selection, while providing more meat with a higher feed efficiency than ever before via intensive farming models. However, in reality, as intensive breeding has continued to grow, the stress response caused by high farming density has followed, especially birds, who are more susceptible to a variety of stressors (such as nutritional, physiological/pathological and environmental) than other domestic species [4]. Oxidative stress and inflammatory response are two of the physiological consequences of such stressors in broilers, which have an intense link and influence each other [5]. This situation may not only affect broiler welfare, but also have adverse effects on antioxidant status and immunity, and increase the risks of facing biological damage, disease threats, and even mortality, which ultimately result in poor performance and economic losses [6,7]. These disruptions are closely associated with excessive production of reactive oxygen species (ROS) [8]. According to statistics, the average annual economic burden induced by the prevalence of disease-causing pathogens and their metabolites pose in the global poultry industry is $3 to $6 billion [9]. Consequently, with the rapid growth of the broiler farming industry and the fact that broilers continuously suffer from some types of stress, it is extremely urgent to supplement an antioxidant to scavenge the harmful ROS and enhance antioxidant function, thus mitigating the adverse effects of stressors.

Antibiotics have been used as feed additives since the 1950s to prevent and control diseases and indirectly accelerate the growth of animals; additionally, they have made a huge contribution to the poultry industry [10]. Nonetheless, it is worth noting that the abuse of antibiotics not only affects the safety of animal products, but also induces the emergence of antibiotic resistant human pathogens and causes the deterioration of the ecological environment [11,12]. Furthermore, following the rise of customers’ attention to health and safety issues, the modern livestock production system has also begun to more consider the concept of clean, green, and ethical animal production practices [13]. Considering these facts, the prohibition of supplementing antibiotics into the diets of livestock has been implemented in many countries of the European Union since 2006, and China has instituted a comprehensive restriction on the use of antibiotics in diets since 2020 as well [14,15]. Thus, it is imperative to study and develop green, economic, and effective alternatives to antibiotic use in poultry feeds. Recently, there is increasing interest in natural products, particularly traditional medicinal plants, as an alternative to antibiotics in feeds. It has been confirmed that many secondary metabolites in the plant, such as flavonoids, play an important role in regulating the antioxidant system and can scavenge ROS, remove oxidation products, and stimulate antioxidant enzymes through antioxidant mechanisms in cell signaling in many animal models, and ultimately, alleviate oxidative stress and enhance the health of animals [16]. Therefore, the herbal extracts are widely applied to protect animals, relieve stress and improve health and performance in Western as well as in many Asian countries, including China and India [16].

*Artemisia ordosica* total flavonoids (ATF) are isolated from *Artemisia ordosica* Krasch (*A. ordosica*, *Compositae* family, *Artemisia* genus), which is a traditional Chinese/Mongolian medicine. *A. ordosica* is a main, representative plant in dry areas of East Asia, especially in the north and northwest of China, such as Inner Mongolia and Xinjiang [17]. The whole plant of *A. ordosica*, even the root, has been utilized as a folk medicine to treat rheumatoid arthritis, cold headache, sore throat, carbuncle, swollen boil, and nasal bleeding, etc. [18]. In addition, owing to their high nutritional value and abundance of flavonoids [19], terpenoids [20], polysaccharides [21], sterols [22], coumarins [23], acetylenes [24], and other bioactive compositions, mainly flavonoid and terpenoid compounds [25,26,27,28,29], *A. ordosica* and their extracts have multiple pharmacological activities, including antimicrobial [30], antioxidative [31], anti-inflammatory [32], immunomodulatory [33], and, therefore, can be efficiently applied in poultry and animal production to promote health. Our previous studies have verified that *A. ordosica* aqueous extracts, mainly comprised of flavonoids, terpenoids, organic acids, and polysaccharides, could effectively improve the growth and antioxidant capacity of broilers and weanling piglets [32,33].

It is now believed that a redox homeostasis imbalance is the major cause of oxidative stress, which is accompanied by an excess generation of ROS and a decline in antioxidant defense systems, and this is considered an important factor in the poor performance and development of various diseases in animals [34]. The endotoxin lipopolysaccharide (LPS), a primary component of the cell wall of gram-negative bacteria, can be recognized by immune cells as a pathogen-associated molecular pattern and consequently induces ROS production, playing a crucial role in the development of oxidative stress, which reduces feed intake and body weight gain, resulting in the animal’s growth being inhibited [35,36]. Intraperitoneal or intravenous injection of LPS can be used to effectively model oxidative stress and inflammatory damage caused by bacterial infection [37]. Our previous study also successfully built an inflammatory immune response model of broilers by injecting LPS and verifying that the *A. ordosica* aqueous extract could effectively alleviate LPS-induced immune over-response and improve growth performance in broilers by lessening the amount of inflammatory cytokines and stress hormone and promoting the growth hormone [32]. However, whether ATF exerts beneficial effects on growth performance and antioxidative capacity in broilers still remains unclear.

Therefore, the present study used ethanol as a solvent to extract the total flavonoids of *A. ordosica* and aimed to investigate the effects of ATF on growth performance, oxidative stress, and antioxidant status in LPS-challenged broilers so as to provide a theoretical basis for the application of *A. ordosica* total flavonoids in poultry production.

## 2. Materials and Methods

### 2.1. Preparation of Total Flavonoids of Artemisia ordosica

Fresh *A. ordosica* (aerial part) was collected from Erdos (Inner Mongolia, China) in July. Raw materials were washed with distilled water and shade-dried at room temperature. The dried materials were smashed and sieved (60 mesh), then the powder was degreased and pigments were removed by petroleum ether in the Soxhlet apparatus for 12 h and dried again at room temperature to reduce petroleum ether residues. ATF was prepared using the method described by Guo et al. [38]. Briefly, after a series of the abovementioned processing, 30 g of dry powder was steeped in 900 mL 60% ethanol aqueous solution (solid:ethanol = 1:30) and ultrasonic-assisted extraction was applied for 1 h at 200 W power, heated reflux at 50 °C, to obtain the extracting solution. The resulting solution was filtered through a 0.45 μm filter and the filtrate was concentrated using a rotary vacuum evaporator (RE-5298, Shanghai Yarong Biochemical Instrument Factory, Shanghai, China) and then lyophilized by a vacuum evaporate to prepare the powder which was stored at −20 °C until use. Total flavonoid content was determined by the spectrophotometric method according to Wang et al. [39], using rutin as the standard and aluminum nitrate as the chromogenic agent. Results were expressed as milligram rutin equivalents (RE)/gram extract and the content of total flavonoid in the extract was 556.1 mg RE/g. These extracts were denominated as ATF and used in the broiler feeding experiment.

### 2.2. Birds, Experimental Design, and Diets

All 240 one-day-old Arbor Acres male broilers were purchased from a local commercial hatchery in Hohhot, Inner Mongolia, China and randomly divided into 5 treatment groups with 6 replicates for each group and 8 broilers in each replicate. Using a completely randomized trial design, the five treatments were as follows: (1) control group (CON), broilers received a basal diet and were treated with 0.9% sterile saline; (2) LPS group (LPS), broilers received a basal diet and underwent LPS challenge; (3) low-dose of the ATF group (ATF-L), broilers received a basal diet supplemented with 500 mg/kg ATF and underwent LPS challenge; (4) middle-dose of the ATF group (ATF-M), broilers received a basal diet supplemented with 750 mg/kg ATF and underwent LPS challenge; (5) high-dose of the ATF group (ATF-H), broilers received a basal diet supplemented with 1000 mg/kg ATF and underwent LPS challenge. According to our previous study, when the ATF level in the basal diet was 750 mg/kg, broilers under normal rearing conditions had better growth performance, immunity, and antioxidant capacity, therefore, the diet containing 750 mg/kg ATF was chosen in the current experiment as a medium dosage treatment group. The experiment lasted for 42 days, divided into the starter phase (day 1 to 15), stress period (day 16 to 28), and convalescence (day 29 to 42). During the stress period (on day 16, 18, 20, 22, 24, 26, and 28), the broilers were injected intra-peritoneally with either LPS solution (*Escherichia coli*, serotype O55:B5, L2880; Sigma-Aldrich, St. Louis, MO, USA) at the dose of 750 μg/kg of body weight (BW) (LPS was dissolved in sterile saline at a concentration of 100 μg/mL) or an equal dose of 0.9% sterile saline.

The feeding trial was conducted on the experimental farm of Inner Mongolia Agricultural University, Hohhot, China. According to the method reported by De Oliveira and Lara [40], the incremental lighting system was adopted in the whole experimental period. The temperature of the experimental room was set at 33 °C for the first 3 days and then gradually reduced by 3 °C every week, reaching a final temperature of 21 °C. The relative humidity and ventilation were maintained at about 50 to 60% and 0.2 to 0.5 m/s, respectively. All broilers were routinely immunized and had ad libitum consumption of diet and water throughout the trial. The vaccination procedure was conducted as follows: the broilers were vaccinated with the Newcastle disease and infectious bronchitis combined vaccine on day 7 and 20, the Newcastle disease, infectious bronchitis, and avian influenza triple vaccine on day 10, and the infectious bursal disease vaccine on day 14 and 24. All diets were fed in mash form and were based on corn−soybean meal formulated to meet or slightly exceed the National Research Council (NRC, 1994) [41] recommendations and nutrients recommendations of Feeding Standard of Chicken, China (NY/T 33-2004) (Chinese Ministry of Agriculture 2004) [42] (Table 1).

### 2.3. Growth Performance Measurement

Broilers were weighed on days 1, 15, 28, and 42, and the feed consumption on each replicate basis was recorded on days 15, 28, and 42 to calculate the average daily gain (ADG), average daily feed intake (ADFI) and feed-to-gain ratios (F/G) for each period.

### 2.4. Sample Collection and Preparation

On days 28 and 42, one broiler was randomly selected from each replicate to collect a blood sample from the wing vein using a vacuum non-anticoagulant tube, and then euthanized by cervical dislocation and dissected to collect the liver and spleen. After standing at room temperature for 30 min, serum sample was detached and collected via centrifugation at 3000× *g* for 15 min at 4 °C and then immediately stored at −20 °C until further analysis. Each tissue sample was rinsed with ice-cold sterile saline solution to remove blood contamination and snap-frozen in liquid nitrogen at first, then stored at −80 °C for the preparation of the homogenate and total RNA isolation.

### 2.5. Assay of Reactive Oxygen Species and Oxidative Byproducts in Serum

Lipid peroxidation, protein, and guanine oxidative damage were assessed by measuring the level of malondialdehyde (MDA), protein carbonyl (PC), and 8-hydroxy-2-deoxyguanosine (8-OHdG) in serum, respectively. Based on the method described by Bai et al. [43], the content of ROS, MDA, PC, and 8-OHdG in serum was determined using commercially available kits supplied by Nanjing Jiancheng Institute of Bioengineering, Nanjing, China, according to the manufacturer’s recommended procedure.

### 2.6. Determination of Antioxidant Indexes in Serum and Tissue Samples

Liver and spleen tissue samples were minced and homogenized with ice-cold saline (wt/vol, 1:9), then centrifuged at 4000× *g* for 15 min at 4 °C. Total antioxidant capacity (TAC), MDA content and the activity of enzymes, including total superoxide dismutase (SOD), glutathione peroxidase (GPx), and catalase (CAT), in serum and tissue homogenate supernatant were measured by spectrophotometric method according to the instructions of the commercial kits (Nanjing Jiancheng Institute of Bioengineering, Nanjing, China). The activity of SOD, GPx, and CAT was expressed as activity unit per milligram of tissue protein (unit/mg protein). The concentration of MDA was expressed as nanomole per milligram of tissue protein (nmol/mg protein). TAC capacity was expressed as micromole (μmol) Trolox equivalent per gram protein of homogenates (μmol/g protein).

### 2.7. Total RNA Extraction and Reverse Transcription

Total RNA from liver and spleen samples (100 mg) was obtained using Trizol reagent (TaKaRa Biotechnology Co., Ltd., Dalian, China). Subsequently, the total RNA was treated with DNase I (TaKaRa) to remove the DNA. The purity and quantity of the total RNA were assessed from the ratio of the absorbance at 260 and 280 nm using a spectrophotometer (Pultton P200CM, San Jose, CA, USA) and adjusted to the same concentration (500 ng/μL) using nuclease-free water. The samples with ratios between 1.8 and 2.1 were considered good quality RNA for cDNA synthesis. RNA integrity was evaluated using horizontal electrophoresis through 1.5% agarose gel. Discrete 18S and 28S rRNA bands were considered as a qualification index for the extracted RNA.

Two microliters (μL) of total RNA (500 ng/μL) from each sample were reverse transcribed to cDNA on LifeECO (TC-96/G/H(b)C, BIOER, Hangzhou, China) using the TB^®^ Green qPCR method with a Prime Script™ RT reagent kit with gDNA Eraser (TaKaRa Biotechnology Co., Ltd., Dalian, China). The reactions were incubated for 15 min at 37 °C, followed by 5 s at 85 °C.

### 2.8. Quantitative Real-Time Polymerase Chain Reaction

Real-time polymerase chain reaction (PCR) was performed using the QuantStudio^®^ 5 real-time PCR Design & Analysis system (LightCycler^®^ 480 II, Roche Diagnostics, Indianapolis, IN, USA) with a TB^®^ Premix Ex Taq™ Kit (Takara Biotechnology Co., Ltd., Dalian, China). Each sample was run in a 20 μL reaction mixture which consisted of 10 μL TB green mix, 0.8 μL each of forward (0.4 μM) and reverse primer (0.4 μM), 2 μL of cDNA, and 6.4 μL of nuclease-free water. The reactions were: 95 °C for 30 s (hold stage), followed by 40 cycles of 95 °C for 5 s, 60 °C for 30 s, and 72 °C for 20 s (PCR stage), then 95 °C for 15 s, 60 °C for 1 min, and 95 °C for 15 s (melt-curve stage). All samples were run in triplicate and melt curve analysis was performed to validate the specificity of the PCR-amplified product. Relative expression levels of target genes were calculated according to the method reported by Pfaffl [44] appropriate for the multi-reference gene normalization procedure. The geometric means of the 3 reference genes, 60S ribosomal protein L13 (*RPL13*) [45], TATA box binding protein (*TBP*) [46], and Tyrosine 3-monooxygenase/tryptophan 5-monooxygenase activation protein zeta polypeptide (*YWHAZ*) [46], were used to normalize the expression data according to the method reported by Hellemans et al. [47]. The PCR primer pair efficiency (E) was calculated from the slope of the standard curve using 5-fold serial dilutions of pooled cDNA samples, according to the equation E= [10^(−1/slope)^ − 1] × 100% [48] (Table 2). The correlation coefficients (*r*^2^) of all standard curves were >0.99 and the amplification efficiency was between 90 and 110%. The specific sequences of primers are listed in Table 2. The sequences of the forward and reverse primers have been described and validated previously [21,45,46,49]. In addition, the specific primers of these genes were verified in silico using the Primer Basic Local Alignment Search Tool (Primer-BLAST, https://www.ncbi.nlm.nih.gov/tools/primer-blast/index.cgi?LINK˙LOC=Blast Home, accessed on 22 September 2022). The melting peaks of the amplification products were determined by the melting curve, which indicated only one expected amplification product had been generated. Each primer pair used yielded a single peak in the melting curve and a single band with the expected size in the 2% agarose gel electrophoresis. Overall, the RT-qPCR investigation complies with the Minimum Information for Publication of Quantitative Real-Time Experiments (MIQE) guidelines [48].

### 2.9. Statistical Analysis

Data obtained are expressed as the mean and standard error of the mean (SEM). The analyses of all data were performed by one-way ANOVA using the GLM procedure of SAS 9.2 (SAS Institute Inc., Cary, NC, USA) with replicates as the experimental unit. Duncan’s multiple range test was used to evaluate differences between the mean values. Here, *p*-values less than 0.05 were considered statistically significant and *p*-values more than or equal to 0.05 were considered statistically non-significant.

## 3. Results

### 3.1. Growth Performance

As shown in Table 3, dietary ATF had no significant effect on growth performance in broilers before the LPS challenge (days 1 to 15) (*p* > 0.05). On day 28 and 42, broiler BW was markedly decreased in the LPS group compared to that in the control group, but was notably alleviated by ATF supplementation, and a dose of 1000 mg/kg had the best alleviating effect (*p* < 0.05). Moreover, compared with the control group, the broilers with LPS intraperitoneal injection alone displayed a significant decrease in ADG and ADFI during the stress period and convalescence (*p* < 0.05). In contrast, the group that was supplemented with 1000 mg/kg ATF exhibited a significant increase in ADG and ADFI compared with the LPS group (*p* < 0.05), and had no significant difference compared to the control group (*p* > 0.05). In the convalescence (days 29 to 42), the ADG of broilers in the ATF-M group (LPS-challenged broilers fed a basal diet supplemented with 750 mg/kg ATF) was higher than that in the LPS group (*p* < 0.05). Over the whole experimental period, there was no significant difference in the feed-to-gain ratio among the groups (*p* > 0.05).

### 3.2. Serum Reactive Oxygen Species and Oxidative Byproducts Concentrations

The effects of ATF on serum ROS, MDA, PC, and 8-OHdG concentrations in broilers challenged with LPS are shown in Table 4. As described in Table 4, on day 28, compared with the CON group, LPS challenge increased the concentrations of ROS, PC, and 8-OHdG in the serum of broilers; however, dietary ATF supplementation obviously suppressed the increased concentrations of the aforementioned indexes of the serum of broilers challenged with LPS (*p* < 0.05). The concentration of ROS and 8-OHdG in all ATF groups was lower than those in the LPS group (*p* < 0.05). In addition, there was no significant difference in ROS concentrations between the ATF-M, ATF-H, and CON groups, nor in 8-OHdG concentrations between the ATF-H and control groups (*p* > 0.05). The concentration of PC in the ATF-M and ATF-H groups was lower than that in the LPS group (*p* < 0.05) but had no difference from the CON group (*p* > 0.05).

The effects of ATF additive and LPS challenge on serum ROS, MDA, PC, and 8-OHdG concentrations in broilers during convalescence are also presented in Table 4. Neither the ATF additive nor the LPS challenge had any effect on the aforementioned indexes of the serum of the broilers on day 42 (*p* > 0.05).

### 3.3. Serum Antioxidant Indexes

As indicated in Table 5, on day 28, compared with the CON group, LPS challenge decreased the levels of CAT, GPx, SOD, and TAC in the serum of broilers; however, dietary ATF supplementation obviously alleviated the decreased levels of the aforementioned indexes of the serum of broilers challenged with LPS (*p* < 0.05). In addition, there was no significant difference in CAT and GPx contents between the control and ATF-related (ATF-L, ATF-M, and ATF-H) groups (*p* > 0.05). The contents of SOD and TAC in the ATF-H group were not significantly different from those in the CON group (*p* > 0.05).

The effects of ATF addition and LPS challenge on serum CAT, GPx, SOD, and TAC levels in broilers during convalescence are also presented in Table 5. Neither the ATF addition nor the LPS challenge had any effect on the aforementioned indexes of the serum of broilers on day 42 (*p* > 0.05).

### 3.4. Hepatic Antioxidant Indexes

As illustrated in Table 6, on day 28, stimulation with LPS alone caused a noticeable decrease of CAT and SOD activities in the liver compared with the control group, whereas ATF addition (500, 750, and 1000 mg/kg) dramatically alleviated the suppression of hepatic CAT and SOD activities in LPS-treated broilers (*p* < 0.05). There was no significant difference in the levels of hepatic CAT, GPx, SOD, TAC, and MDA among the groups on day 42 (*p* > 0.05).

### 3.5. Hepatic Antioxidant-Related Gene Expression Levels

To determine whether the antioxidation action of ATF was mediated through the Kelch-like ECH-associated protein 1 (Keap1)/nuclear factor erythroid 2-related factor 2 (Nrf2) signaling pathway, the gene expression of *Keap1*, *Nrf2*, *SOD*, *CAT*, and *GPx* was assessed by quantitative real-time PCR analysis. As summarized in Figure 1, on day 28, compared to the control group, LPS administration decreased the gene expression of *SOD*, *CAT*, and *Nrf2*, but increased the gene expression of *Keap1* in the liver tissue of broilers; however, the negative changes in the mRNA expression of the aforementioned genes were markedly reversed by adding ATF in diets, except for *Nrf2*, which increased significantly only with 750 and 1000 mg/kg ATF supplementation (*p* < 0.05). No significant difference in the expression of these aforementioned genes was observed among the groups on day 42. (*p* > 0.05).

### 3.6. Splenic Antioxidant Indexes

As illustrated in Table 7, on day 28, the broilers that were injected with LPS only exhibited a marked increase in MDA and a decrease in GPx and SOD in spleen compared with the control group, whereas ATF supplementation markedly reversed the negative changes in the aforementioned indexes (*p* < 0.05). The activity of GPx in all ATF groups was higher than that in the LPS group (*p* < 0.05), and there was no significant difference among the ATF-M, ATF-H, and CON groups (*p* > 0.05). The activity of SOD in the ATF-H group was higher than that in the LPS group (*p* < 0.05) but had no difference from the CON group (*p* > 0.05). The content of MDA in all ATF groups was lower than that in the LPS group (*p* < 0.05), and there was no significant difference between the ATF-H group and the CON group (*p* > 0.05). On day 42, the splenic MDA content of the ATF-M and ATF-H groups was lower than that in the LPS group (*p* < 0.05), but had no difference from the CON group (*p* > 0.05).

### 3.7. Splenic Antioxidant-Related Gene Expression Levels

The splenic gene expression data from the broilers are shown in Figure 2. On day 28, LPS markedly decreased the gene expression of *SOD*, *CAT*, *GPx*, and *Nrf2*, whereas it elevated the gene expression of *Keap1* in the spleen tissue of broilers (*p* < 0.05); however, dietary ATF supplementation obviously alleviated negative changes induced by LPS, and the ATF-H group had the best alleviating effect (*p* < 0.05).

On day 42, the mRNA expression of splenic *GPx* and *Nrf2* in the LPS group was significantly lower than that in the CON, ATF-M, and ATF-H groups (*p* < 0.05). The splenic *Keap1* mRNA expression in the LPS group was significantly higher than that in the other groups (*p* < 0.05).

## 4. Discussion

The study of safe and environmentally friendly herbal extracts has recently gained much attention on account of their potent bioactivities, such as antioxidant and anti-inflammatory effects. Nonetheless, how different extracts from different medicinal plants regulate the balance between ROS and antioxidants triggered during oxidative stress remains to be further investigated. In the present experiment, we applied a classical *E. coli* LPS-stimulated model to induce oxidative injury in broilers by intraperitoneal administration of LPS and evaluated the effect of ATF on an LPS-induced oxidative stress broiler model.

In the current study, we found that LPS challenge decreased the BW and ADG in both the stress period and convalescence compared to those in the unchallenged broilers, which was possibly due to simultaneously compromised ADFI. This is consistent with the findings of Li et al. [32] and Yang et al. [50], where LPS-induced growth retardation was reported in broilers exposed to 500 μg/kg BW LPS. This phenomenon could be linked to the diversion of dietary nutrients from growth to support inflammatory-related processes, such as the synthesis of cytokines and various acute proteins [51]. Another possible reason for the observed phenomenon could be the ability of the overproduction of inflammatory cytokines to lose appetite via the control of the hypothalamic−pituitary−adrenal axis-associated antioxidant process, which was reflected in the decrease of ADFI in the LPS group broilers in the present study [52]. We also observed that dietary ATF alleviated the decreased BW (all three ATF treatments), as well as improved the ADG and ADFI (1000 mg/kg ATF treatment) in LPS-challenged broilers during the stress period and convalescence, thereby alleviating the inhibition of the growth performance of broilers caused by LPS challenge. These benefits of growth in infected broilers were possibly attributed to the antioxidant characteristics of flavonoids. Xing et al. [33] reported that flavonoids of *A. ordosica* aqueous extracts might contribute to the positive effects on growth performance by improving the apparent nutrient digestibility of weanling piglets in a dose-dependent manner. Yang et al. [50] also reported that *A. argyi* flavonoids had better potential to improve the growth performance of broilers under inflammation. These findings are consistent with the results of the current study. Furthermore, another study indicated that LPS and *A. ordosica* extract (rich in flavonoid and phenolic) exhibited an interaction related to the content of insulin-like growth factor-1 (IGF-1) in serum; in other words, *A. ordosica* extract could also relieve the growth inhibition of broilers challenged with LPS by promoting the secretion of growth-promoting hormones [32]. More specifically, it has been proven that flavonoids can promote the combination of growth hormone and hepatic growth hormone receptor, thereby inducing growth promotion [53]. These results may also partially explain our findings.

It has been proven that LPS can induce microglial activation and increase the production of ROS in mitochondria, which causes redox imbalance, producing oxidative stress that affects the health and growth of organisms [54]. Excessive ROS, including superoxide ion (O_2_^•^), hydroxyl radical (HO^•^), and hydrogen peroxide (H_2_O_2_), are very reactive molecules that can react with metal ions (Fe^2+^) and directly cause cell and tissue oxidative damage via the formation of DNA adducts, lipid peroxidation, and protein cross-linking [55]. On this basis, in addition to ROS, the measurement of oxidative byproduct levels in the serum, including MDA (a biomarker for lipid peroxidation), 8-OHdG (a biomarker of oxidative DNA damage), and PC (a biomarker of oxidative protein damage), is also applied to assess the load of oxidative stress in the current study. Our results showed that the LPS challenge caused an overproduction of serum ROS, 8-OHdG, and PC in broilers during the stress period, which suggested that LPS disturbed redox homeostasis and contributed to oxidative injury due to ROS accumulation. Interestingly, we observed a lowered concentration of blood metabolites (ROS, 8-OHdG, and PC) in the serum of LPS-challenged broilers supplemented with ATF. This may be related to the fact that flavonoids with enormous antioxidant activity may help neutralize excessive ROS and reduce oxidative stress. Qi et al. [26] assessed the antioxidant properties of four *A. ordosica* flavonoids, including genkwanin, hydroxygenkwanin, apigenin-7,4′-dimethylether, and rhamnetin, through the scavenging activities of hydroxyl radicals, which suggested that *A. ordosica* flavonoids exerted the ability to scavenge radicals in vitro. Li et al. [56] reported that the antioxidant property of natural flavonoids from *A. ordosica* against superoxide anion radicals might be attributed to its structure, in which the phenolic hydroxyl groups could quench singlet oxygens and/or trap other free radicals. Consistent with our results, another study indicated that orange peel extract (rich in proanthocyanidins and other flavonoids) reduced H_2_O_2_ production in the muscles of broiler chickens under heat stress [57]. In fact, as a class of essential metabolites of physiological processes, ROS are commonly maintained at a specific level by a sophisticated endogenous antioxidant defense system, including both enzymatic (SOD, GPx, CAT and so on) and non-enzymatic (ascorbic acid, *α*-tocopherol, *β*-carotene and so on) scavengers, which activate specific transcription factors and regulate a variety of biochemical pathways as signal molecules [58,59,60]. The results of this study on the excessive elevation of serum oxidative byproducts might reflect an overwhelming of the antioxidant defense system due to the increased generation of ROS caused by LPS. The SOD, CAT, GPx, and TAC levels are considered to be four reliable indicators reflecting a broiler’s antioxidant status, and they can also act as biomarkers of oxidative stress. In the current study, the broilers with LPS instillation alone observed a marked decrease in serum SOD, CAT, GPx, and TAC when compared with those who received saline injections, indicating that LPS stress has a lower antioxidant and enzymatic defense, which may be attributed to a direct inhibitory oxidative effect and/or their depletion by the antioxidant response, due to the presence of high ROS levels [61]. However, when ATF was previously and concomitantly administered with LPS, it significantly alleviated the inhibitory effect on antioxidant enzyme activity and total antioxidant capacity in the serum of broilers challenged with LPS. Consistent with our results, Dhibi et al. [62] also found that *Artemisia arborescens* hydroalcoholic extract (rich in flavonoids, including quercetin, rutin, luteolin, kaempferol, and isorhamnetin) effectively prevented the kidney from experiencing oestroprogestative-induced oxidative damage in rats by returning SOD, CAT, and GPx activities to normal levels. Combined with these results and based on our study, it is reasonable to speculate that ATF can not only exert direct antioxidant effects acting as free-radical scavengers and singlet oxygen quenchers, directly causing low ROS levels, but also enhance enzymatic antioxidant activities, successfully eliminating ROS, PC, and 8-OHdG generation, thereby protecting from oxidative stress damage. It is worth noting that, throughout the study, no significant difference was observed in the content of serum MDA among all groups. This can be explained by the short half-life of MDA, which contributes to a rapid return to the baseline, but can also be due to the data on serum MDA, which is not a good representation of damage because the content of MDA may be diluted when compared with the content in injured organs caused by LPS [63,64].

To gain further insight into the mitigating effects of ATF against LPS-induced oxidative stress in broilers, we selected the liver and spleen as the target organs. The related pathological condition of the liver and spleen is mainly caused by oxidative stress, which forces the development of similar changes in other visceral organs [65]. In addition, many bioactive compounds, such as flavonoids and polysaccharides, can reach the liver and exert the potential beneficial effect of antioxidants via the portal vein after absorption along the gastrointestinal tract [66,67]. As expected, the antioxidase activities were decreased in the liver (CAT and SOD) and spleen (GPx and SOD) of broilers by LPS induction during the stress period; however, ATF administration effectively alleviated the loss of these antioxidase activities after the LPS challenge. Similarly, the gene expression of *CAT*, *SOD*, and *GPx* also showed the same variations, but this might partially explain the increase in the activity of the antioxidases because the variations in the gene expression of splenic *CAT* on day 28 and splenic *GPx* on day 42 were not completely consistent with the variations in enzymatic activities. The results might be due, in part, to the fact that the enzymatic activity was modulated by multiple factors rather than merely by gene expression [68]. Moreover, our study indicated that dietary ATF supplementation was beneficial for protecting tissues from LPS-induced lipid peroxidation, as evidenced by the enhanced antioxidase activities causing the reduced production of MDA, which ultimately alleviated the oxidative stress of broilers. Meanwhile, we noticed that the LPS group’s hepatic MDA level returned to normal during convalescence, but the high level of splenic MDA continued throughout convalescence, suggesting that the spleen recovers more slowly after oxidative stress injury, whereas the recovery process could be accelerated by the dietary addition of ATF.

The redox-sensitive Nrf2 responds to oxidative stress and can regulate the expression of downstream representative target genes involved in the antioxidation process, such as *SOD*, *CAT*, and *GPx*, which have a significant effect on the elimination of excessive ROS [69]. Under physiological conditions, Nrf2 binds to its inhibitor Keap1 in the cytoplasm and acts in an inactive form. However, at the occurrence of oxidative stress, Keap1 is modified to release Nrf2 from the cytoplasm into the nucleus, which initiates the transcription of downstream antioxidant protective genes [70]. Numerous studies have suggested that Nrf2 is involved in the antioxidative activity of flavonoids isolated from a variety of plants. As an extension, we deduced that ATF has a similar antioxidative effect against oxidative stress caused by LPS challenge and primarily explored a possible action mechanism by detecting the mRNA expressions of five candidate key nodes in the Keap1/Nrf2 signaling pathways in the liver and spleen, including *Keap1*, *Nrf2*, *SOD*, *CAT*, and *GPx*. In the present study, the results showed that when broilers are exposed to LPS, ATF attenuated the inhibition of *SOD*, *CAT*, and *GPx* gene expression. In addition, ATF could upregulate the *Nrf2* gene expression, but downregulate the *Keap1* gene expression. The downregulation of *Keap1* gene expression may facilitate the activation of *Nrf2*, which is further translocated into the nucleus and binds to the antioxidant responsive element [21]. This view was further supported by the increased gene expression level of hepatic and splenic *Nrf2*. Our results proved that treatment with ATF could induce antioxidant gene expression and increase antioxidant enzyme activity for protection against oxidative stress caused by LPS. This finding is consistent with a previous study in which improved antioxidant capacity was demonstrated in heat-stressed broilers fed a diet supplemented with enzymatically (cellulase and pectinase) treated *Artemisia annua* (rich in flavonoids and phenolics) [71]. Flavonoids isolated from *A. ordosica* have been confirmed in vitro in both antioxidant and anti-inflammatory aspects [19,26]. Additionally, as the main component in ATF was total flavonoids (556.1 mg rutin equivalents/g ATF), it is reasonable to speculate that ATF may have the same antioxidant effect as rutin which has been proven to protect LPS-induced oxidative stress in a murine model [72]. Based on the results of our study, we preliminarily speculated that ATF could ameliorate oxidative stress and might be related to the mediation of Nrf2 pathways. However, the specific mechanism of action still needs further verification. It is worth noting that the presence of active constituents in herbal extracts can simultaneously act on different targets in the antioxidant-related pathway [73]. Moreover, increasing evidence demonstrates that the excess production of ROS is usually associated with high levels of proinflammatory cytokines, suggesting that crosstalk between the Nrf2 and other pathways, including nuclear factor kappa B (NF-κB), may be an important regulatory mechanism in many cellular responses to various stresses, such as oxidative stress and immunological stress.

Collectively, *A. ordosica* has a great deal of advantages, such as abundant resources, low price, rich nutritional value, and high pharmacological activity, and possesses great application and development value. Unfortunately, as far as we know, studies on the effects of *A. ordosica* flavonoids as a feed additive on the growth performance and health of livestock are limited, especially in poultry. Consequently, further studies of different in vitro and in vivo disease models should be carried out to completely evaluate the role and the exact mechanism of ATF as an antioxidant and immunomodulatory agent.

## 5. Conclusions

In conclusion, the present study reported for the first time that ATF protected against LPS-induced oxidative stress in an animal model. The results obtained from this study showed that LPS injection inhibited the growth performance of broilers and led to oxidative stress, which manifested as high levels of ROS and oxidative byproducts (MDA, PC, and 8-OHdG) in the serum and low levels of antioxidant enzyme activity (SOD, CAT, and GPx) in the serum and tissues. However, dietary ATF supplementation could reverse these negative effects and alleviate oxidative stress, thereby improving the growth performance of LPS-challenged broilers. Under the conditions of this experiment, the recommended level of ATF in the broiler diet was 1000 mg/kg.

## Figures and Tables

**Figure 1 antioxidants-11-01985-f001:**
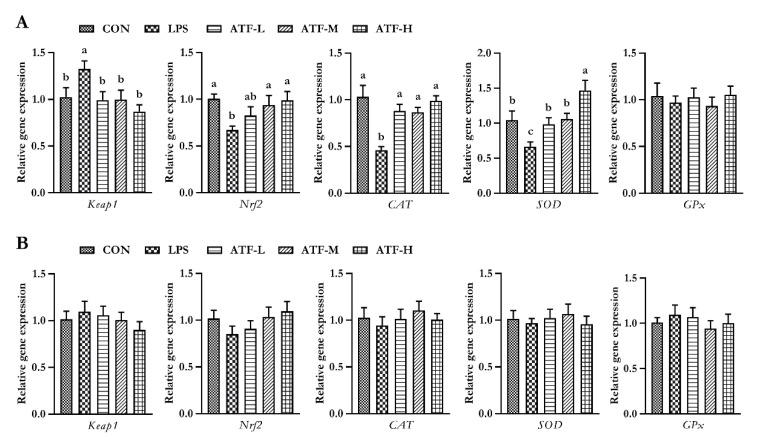
Effect of ATF on the expression of antioxidant-related genes in the liver of broilers challenged with LPS. (**A**) The data of antioxidant-related gene expression in the liver of broilers at 28 days. (**B**) The data of antioxidant-related gene expression in the liver of broilers at 42 days. Abbreviations: Keap1, Kelch-like ECH-associated protein-1; Nrf2, nuclear factor-erythroid 2-related factor 2; CAT, catalase; SOD, total superoxide dismutase; GPx, glutathione peroxidase; ATF, *Artemisia ordosica* alcohol extract; CON, non-challenged broilers fed a basal diet; LPS, lipopolysaccharide-challenged broilers fed a basal diet; ATF-L, lipopolysaccharide-challenged broilers fed a basal diet supplemented with 500 mg/kg ATF; ATF-M, lipopolysaccharide-challenged broilers fed a basal diet supplemented with 750 mg/kg ATF; ATF-H, lipopolysaccharide-challenged broilers fed a basal diet supplemented with 1000 mg/kg ATF. Each value is shown as the mean and standard error of mean (SEM) (*n* = 6); Bars with different letters (a, b, and c) are significantly different (*p* < 0.05), while bars with the same letter indicate no significant difference between groups (*p* ≥ 0.05).

**Figure 2 antioxidants-11-01985-f002:**
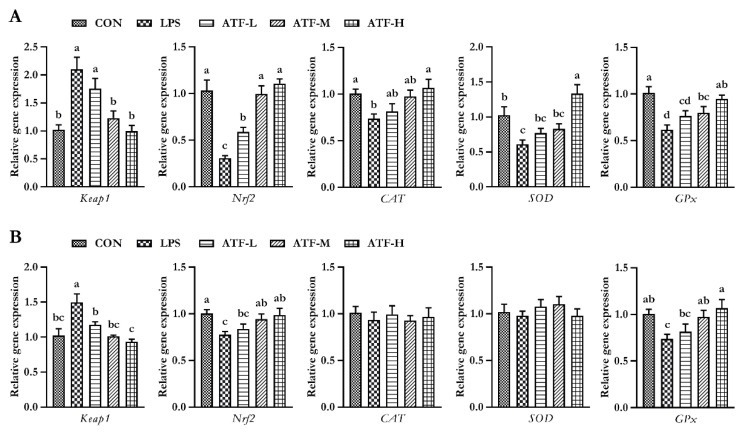
Effect of ATF on the expression of antioxidant-related genes in the spleen of broilers challenged with LPS. (**A**) The data of antioxidant-related gene expression in the spleen of broilers at 28 days. (**B**) The data of antioxidant-related gene expression in the spleen of broilers at 42 days. Abbreviations: Keap1, Kelch-like ECH-associated protein-1; Nrf2, nuclear factor-erythroid 2-related factor 2; CAT, catalase; SOD, total superoxide dismutase; GPx, glutathione peroxidase; ATF, *Artemisia ordosica* alcohol extract; CON, non-challenged broilers fed a basal diet; LPS, lipopolysaccharide-challenged broilers fed a basal diet; ATF-L, lipopolysaccharide-challenged broilers fed a basal diet supplemented with 500 mg/kg ATF; ATF-M, lipopolysaccharide-challenged broilers fed a basal diet supplemented with 750 mg/kg ATF; ATF-H, lipopolysaccharide-challenged broilers fed a basal diet supplemented with 1000 mg/kg ATF. Each value is shown as the mean and standard error of mean (SEM) (*n* = 6); Bars with different letters (a, b, c, and d) are significantly different (*p* < 0.05), while bars with the same letter indicate no significant difference between groups (*p* ≥ 0.05).

**Table 1 antioxidants-11-01985-t001:** Composition and nutrient levels of the basal diet (as-fed basis).

Items	1 to 21 Days of Age	22 to 42 Days of Age
Ingredients, %		
Corn	52.50	58.80
Soybean meal	40.00	33.80
Soybean oil	3.00	3.00
Dicalcium phosphate	1.90	1.80
Limestone	1.08	1.22
Salt	0.37	0.37
Lysine	0.05	0.03
Methionine	0.19	0.07
Premix ^1^	0.80	0.80
Choline chloride	0.11	0.11
Total	100.00	100.00
Nutrient levels ^2^		
Metabolic energy, MJ/kg	12.42	12.62
Crude protein	21.77	19.65
Calcium	1.00	1.02
Available phosphorus	0.44	0.42
Lysine	1.34	1.15
Methionine	0.55	0.40
Cystine	0.40	0.36

^1^ Premix provided the following per kilogram of diet: vitamin A 9000 IU, vitamin D_3_ 3000 IU, vitamin E 26 mg, vitamin K_3_ 1.20 mg, vitamin B_1_ 3.00 mg, vitamin B_2_ 8.00 mg, vitamin B_6_ 4.40 mg, vitamin B_12_ 0.012 mg, nicotinic acid 45 mg, folic acid 0.75 mg, biotin 0.20 mg, calcium pantothenate 15 mg, Fe 100 mg, Cu 10 mg, Zn 108 mg, Mn 120 mg, I 1.5 mg, Se 0.35 mg. ^2^ Crude protein was a measured value, while others were all calculated values.

**Table 2 antioxidants-11-01985-t002:** Primer sequences and parameters.

Genes	GenBank Accession No.	Primer Sequences, 5′–3′	Length, bp	Efficiency, %
*Keap1*	KU321503.1	F-CTGCTGGAGTTCGCCTACAC	96	104.85
		R-CACGCTGTCGATCTGGTACA		
*Nrf2*	NM_205117.1	F-GATGTCACCCTGCCCTTAG	215	102.53
		R-CTGCCACCATGTTATTCC		
*CAT*	NM_001031215.1	F-GTTGGCGGTAGGAGTCTGGTCT	182	101.14
		R-GTGGTCAAGGCATCTGGCTTCTG		
*SOD*	NM_205064.1	F-TTGTCTGATGGAGATCATGGCTTC	98	100.19
		R-TGCTTGCCTTCAGGATTAAAGTGA		
*GPx*	NM_001163245.1	F-CAAAGTTGCGGTCAGTGGA	136	102.01
		R-AGAGTCCCAGGCCTTTACTACTTTC		
*RPL13*	NM_204999.1	F-GGAGGAGAAGAACTTCAAGGC	66	104.32
		R-CCAAAGAGACGAGCGTTTG		
*TBP*	NM_205103.1	F-CCGGAATCATGGATCAGAAC	85	103.23
		R-GGAATTCCAGGAGTCATTGC		
*YWHAZ*	NM_001031343.2	F-TTGCTGCTGGAGATGACAAG	61	104.17
		R-CTTCTTGATACGCCTGTTG		

Abbreviations: Keap1, Kelch-like ECH-associated protein-1; Nrf2, nuclear factor-erythroid 2-related factor 2; CAT, catalase; SOD, total superoxide dismutase; GPx, glutathione peroxidase; RPL13, 60S ribosomal protein L13; TBP, TATA box binding protein; YWHAZ, tyrosine 3-monooxygenase/tryptophan 5-monooxygenase activation protein zeta polypeptide; F, forward primer; R, reverse primer.

**Table 3 antioxidants-11-01985-t003:** Effect of ATF on the growth performance of broilers challenged with LPS.

Items	Treatments					SEM	*p*-Value
	CON	LPS	ATF-L	ATF-M	ATF-H		
BW, g							
15 d	434.81	428.33	444.91	455.48	465.62	12.92	0.246
28 d	1193.37 ^a^	1068.19 ^c^	1123.28 ^b^	1133.56 ^b^	1193.57 ^a^	25.90	<0.001
42 d	2393.57 ^ab^	2092.52 ^d^	2231.86 ^c^	2314.33 ^bc^	2452.28 ^a^	65.22	<0.001
ADG, g/d							
1 to 15 d	26.45	26.01	27.12	27.83	28.50	0.86	0.251
16 to 28 d	58.35 ^a^	49.22 ^c^	52.18 ^bc^	52.16 ^bc^	56.00 ^ab^	2.13	0.012
29 to 42 d	85.73 ^ab^	73.17 ^c^	79.19 ^bc^	84.34 ^ab^	89.91 ^a^	3.65	0.005
ADFI, g/d							
1 to 15 d	36.08	36.37	35.24	36.99	37.91	1.10	0.532
16 to 28 d	92.89 ^a^	80.30 ^c^	81.91 ^bc^	84.45 ^abc^	90.13 ^ab^	3.52	0.038
29 to 42 d	160.37 ^a^	142.97 ^b^	155.55 ^ab^	155.05 ^ab^	164.90 ^a^	5.16	0.025
F/G							
1 to 15 d	1.37	1.40	1.30	1.33	1.33	0.03	0.219
16 to 28 d	1.60	1.64	1.58	1.63	1.61	0.06	0.980
29 to 42 d	1.88	1.96	1.97	1.84	1.84	0.06	0.361

Abbreviations: BW, body weight; ADG, average daily gain; ADFI, average daily feed intake; F/G, feed-to-gain ratio; ATF, *Artemisia ordosica* alcohol extract; CON, non-challenged broilers fed a basal diet; LPS, lipopolysaccharide-challenged broilers fed a basal diet; ATF-L, lipopolysaccharide-challenged broilers fed a basal diet supplemented with 500 mg/kg ATF; ATF-M, lipopolysaccharide-challenged broilers fed a basal diet supplemented with 750 mg/kg ATF; ATF-H, lipopolysaccharide-challenged broilers fed a basal diet supplemented with 1000 mg/kg ATF. Each value is shown as the mean and standard error of mean (SEM) (*n* = 6); Different superscript letters (a, b, c, and d) within the same row indicate significant differences between experimental groups (*p* < 0.05), while the same letter indicates no significant difference between groups (*p* ≥ 0.05).

**Table 4 antioxidants-11-01985-t004:** Effect of ATF on serum ROS and oxidative byproducts in broilers challenged with LPS.

Items	Treatments					SEM	*p*-Value
	CON	LPS	ATF-L	ATF-M	ATF-H		
28 d							
ROS, IU/mL	114.67 ^c^	403.65 ^a^	374.13 ^b^	115.88 ^c^	112.82 ^c^	56.10	<0.001
MDA, nmol/mL	3.07	3.43	3.48	3.19	3.14	0.18	0.436
PC, nmol/mg prot.	1.08 ^b^	2.25 ^a^	2.18 ^a^	1.10 ^b^	1.15 ^b^	0.23	<0.001
8-OHdG, ng/mL	3.42 ^d^	15.91 ^a^	13.09 ^b^	8.29 ^c^	3.35 ^d^	2.10	<0.001
42 d							
ROS, IU/mL	92.68	99.70	91.45	93.71	92.89	3.81	0.605
MDA, nmol/mL	3.64	3.76	3.63	3.78	3.51	0.16	0.792
PC, nmol/mg prot.	1.14	1.23	1.17	1.22	1.19	0.03	0.164
8-OHdG, ng/mL	3.14	3.34	3.12	3.19	3.13	0.17	0.890

Abbreviations: ROS, reactive oxygen species; MDA, malondialdehyde; PC, protein carbonyl; 8-OHdG, 8-hydroxy-2-deoxyguanosine; ATF, *Artemisia ordosica* alcohol extract; CON, non-challenged broilers fed a basal diet; LPS, lipopolysaccharide-challenged broilers fed a basal diet; ATF-L, lipopolysaccharide-challenged broilers fed a basal diet supplemented with 500 mg/kg ATF; ATF-M, lipopolysaccharide-challenged broilers fed a basal diet supplemented with 750 mg/kg ATF; ATF-H, lipopolysaccharide-challenged broilers fed a basal diet supplemented with 1000 mg/kg ATF. Each value is shown as the mean and standard error of mean (SEM) (*n* = 6); Different superscript letters (a, b, c, and d) within the same row indicate significant differences between experimental groups (*p* < 0.05), while the same letter indicates no significant difference between groups (*p* ≥ 0.05).

**Table 5 antioxidants-11-01985-t005:** Effect of ATF on serum antioxidant indexes in broilers challenged with LPS.

Items	Treatments					SEM	*p*-Value
	CON	LPS	ATF-L	ATF-M	ATF-H		
28 d							
CAT, U/mL	3.11 ^a^	2.33 ^b^	2.88 ^a^	3.06 ^a^	3.09 ^a^	0.16	<0.001
SOD, U/mL	390.59 ^a^	196.25 ^d^	250.85 ^c^	332.94 ^b^	375.70 ^ab^	33.89	<0.001
GPx, U/mL	2750.88 ^ab^	1960.23 ^c^	2392.98 ^b^	2734.50 ^ab^	3033.32 ^a^	187.69	<0.001
TAC, mM	0.64 ^a^	0.04 ^c^	0.20 ^b^	0.59 ^a^	0.62 ^a^	0.10	<0.001
42 d							
CAT, U/mL	5.30	4.97	5.40	5.13	5.53	0.22	0.422
SOD, U/mL	436.37	437.59	430.83	432.06	436.37	19.31	0.999
GPx, U/mL	2747.29	2525.58	2510.08	2671.32	2820.16	133.54	0.406
TAC, mM	0.80	0.76	0.76	0.80	0.85	0.03	0.265

Abbreviations: CAT, catalase; SOD, total superoxide dismutase; GPx, glutathione peroxidase; TAC, total antioxidant capacity; ATF, *Artemisia ordosica* alcohol extract; CON, non-challenged broilers fed a basal diet; LPS, lipopolysaccharide-challenged broilers fed a basal diet; ATF-L, lipopolysaccharide-challenged broilers fed a basal diet supplemented with 500 mg/kg ATF; ATF-M, lipopolysaccharide-challenged broilers fed a basal diet supplemented with 750 mg/kg ATF; ATF-H, lipopolysaccharide-challenged broilers fed a basal diet supplemented with 1000 mg/kg ATF. Each value is shown as the mean and standard error of mean (SEM) (*n* = 6); Different superscript letters (a, b, c, and d) within the same row indicate significant differences between experimental groups (*p* < 0.05), while the same letter indicates no significant difference between groups (*p* ≥ 0.05).

**Table 6 antioxidants-11-01985-t006:** Effect of ATF on hepatic antioxidant indexes in broilers challenged with LPS.

Items	Treatments					SEM	*p*-Value
	CON	LPS	ATF-L	ATF-M	ATF-H		
28 d							
CAT, U/mg prot.	10.16 ^a^	4.83 ^b^	9.33 ^a^	9.15 ^a^	9.20 ^a^	0.92	<0.001
SOD, U/mg prot.	851.64 ^a^	663.72 ^b^	883.67 ^a^	868.69 ^a^	874.5 ^a^	57.16	0.020
GPx, U/mg prot.	42.82	48.60	43.63	49.07	49.14	2.67	0.238
TAC, μmol/g prot.	84.08	71.74	74.71	75.71	88.72	5.13	0.092
MDA, nmol/mg prot.	0.43	0.48	0.45	0.41	0.45	0.04	0.782
42 d							
CAT, U/mg prot.	18.28	18.26	19.47	19.18	20.25	1.39	0.852
SOD, U/mg prot.	1464.90	1555.64	1680.24	1564.21	1471.64	114.69	0.705
GPx, U/mg prot.	94.21	91.10	97.69	92.53	94.05	5.94	0.961
TAC, μmol/g prot.	103.73	92.78	90.94	95.73	103.09	4.99	0.238
MDA, nmol/mg prot.	0.53	0.58	0.54	0.54	0.49	0.05	0.803

Abbreviations: CAT, catalase; SOD, total superoxide dismutase; GPx, glutathione peroxidase; TAC, total antioxidant capacity; MDA, malondialdehyde; ATF, *Artemisia ordosica* alcohol extract; CON, non-challenged broilers fed a basal diet; LPS, lipopolysaccharide-challenged broilers fed a basal diet; ATF-L, lipopolysaccharide-challenged broilers fed a basal diet supplemented with 500 mg/kg ATF; ATF-M, lipopolysaccharide-challenged broilers fed a basal diet supplemented with 750 mg/kg ATF; ATF-H, lipopolysaccharide-challenged broilers fed a basal diet supplemented with 1000 mg/kg ATF. Each value is shown as the mean and standard error of mean (SEM) (*n* = 6); Different superscript letters (a and b) within the same row indicate significant differences between experimental groups (*p* < 0.05), while the same letter indicates no significant difference between groups (*p* ≥ 0.05).

**Table 7 antioxidants-11-01985-t007:** Effect of ATF on splenic antioxidant indexes in broilers challenged with LPS.

Items	Treatments					SEM	*p*-Value
	CON	LPS	ATF-L	ATF-M	ATF-H		
28 d							
CAT, U/mg prot.	0.59	0.44	0.54	0.57	0.59	0.05	0.168
SOD, U/mg prot.	170.72 ^ab^	118.75 ^c^	138.31 ^bc^	153.31 ^abc^	187.48 ^a^	14.76	0.003
GPx, U/mg prot.	111.82 ^a^	72.61 ^c^	90.65 ^b^	100.85 ^ab^	107.86 ^a^	7.75	<0.001
TAC, μmol/g prot.	86.75	78.11	82.37	79.24	85.11	7.29	0.919
MDA, nmol/mg prot.	1.10 ^c^	3.12 ^a^	1.72 ^b^	1.86 ^b^	1.30 ^c^	0.32	<0.001
42 d							
CAT, U/mg prot.	0.40	0.44	0.49	0.51	0.46	0.04	0.387
SOD, U/mg prot.	145.63	155.64	162.77	142.54	163.66	13.44	0.750
GPx, U/mg prot.	109.95	88.51	89.81	98.51	98.66	6.71	0.154
TAC, μmol/g prot.	87.54	84.60	82.54	82.56	81.98	4.77	0.932
MDA, nmol/mg prot.	1.11 ^c^	1.77 ^a^	1.55 ^ab^	1.24 ^bc^	1.29 ^bc^	0.15	0.008

Abbreviations: CAT, catalase; SOD, total superoxide dismutase; GPx, glutathione peroxidase; TAC, total antioxidant capacity; MDA, malondialdehyde; ATF, *Artemisia ordosica* alcohol extract; CON, non-challenged broilers fed a basal diet; LPS, lipopolysaccharide-challenged broilers fed a basal diet; ATF-L, lipopolysaccharide-challenged broilers fed a basal diet supplemented with 500 mg/kg ATF; ATF-M, lipopolysaccharide-challenged broilers fed a basal diet supplemented with 750 mg/kg ATF; ATF-H, lipopolysaccharide-challenged broilers fed a basal diet supplemented with 1000 mg/kg ATF. Each value is shown as the mean and standard error of mean (SEM) (*n* = 6); Different superscript letters (a, b, and c) within the same row indicate significant differences between experimental groups (*p* < 0.05), while the same letter indicates no significant difference between groups (*p* ≥ 0.05).

## Data Availability

The data presented in this study are available upon request from the corresponding author.

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
