# Peer review of "Effects of Total Flavonoids of Artemisia ordosica on Growth Performance, Oxidative Stress, and Antioxidant Status of Lipopolysaccharide-Challenged Broilers"

_antioxidants, 2022, doi:10.3390/antiox11101985_

Round 1
Reviewer 1 Report
Dear authors,
this is a very well planned study and will interest many readers, because many researchers are involved in the investigation of flavonoids even in veterinary medicine.
I recommend publication of this manuscript after minor revision.
Pages 6-7, Table 3: insert lines before ADG g/d and ADFI g/d in the table for better understanding.
Page 7, lines 267-272: rephrase "were no significant differences"
Page 8, lines 287-296: The content is redundant. Please shorten and summarize.
Page 10, line 348 and page 11, line 391: Where is the SEM seen in the figure?
Page 12, line 475: Change "present" to "presence"
Page 14, line 552: insert "and" between "stress and might"
Page 14, line 567: change "in" to "as antioxidant and immunomodulatory agent."
Reviewer 2 Report
Effects of Total Flavonoids of Artemisia ordosica on Growth Performance, Oxidative Stress, and Antioxidant Status of Lipopolysaccharide-Challenged Broilers
LPS, an endotoxin from gram negative bacterial induces inflammatory cytokines and stress hormones that has a negative impact on poultry growth and oxidative stress. Previous studies reported that an extract from A. ordosica alleviated LPS-induced immu ne overresponse and improve growth performance in broilers. The authors proposed to perform a similar study to evaluate the effect of the extract administered to broilers on growth performance and antioxidative capacity.
Although the introduction appears to suggest that the authors are reperforming the same experiment as previously published, the study really focused on investigating the molecular mechanisms of effect of the extract on specific antioxidant gene transcription.
Although the study was adequately designed, there are some issues that need to be addressed prior to publication.
Specifics comments
Table 2. XM_015274015.1 is a partial mRNA for mucin 2-like. In addition, NCBI reports that "This record was removed as a result of standard genome annotation processing." Please resolve.
The most concerning issue is with the RNA isolation and PCR techniques. The details of both methods described between lines 207 – 226, which are completely inadequate. Without having confidence in the methods it is impossible to evaluate the data resulting from those methods.
In short RNA was not evaluated for quality by current standard methods, the primers were not evaluated for their use in qRT-PCR and the use of a single housekeeping gene is not acceptable.
The following references need to be placed in the context of the document.
The major concerns with the quantitative PCR are:
1. that there is a need to determine the efficiency of the primers used for qPCR.
A powerful way to determine whether your qPCR assay is optimized is to run serial dilutions of a template and use the results to generate a standard curve. The template used for this purpose can be a target with known concentration (e.g., nanograms of genomic DNA or copies of plasmid DNA) or a sample of unknown quantity (e.g., cDNA). The standard curve is constructed by plotting the log of the starting quantity of template (or the dilution factor, for unknown quantities) against the CT value obtained during amplification of each dilution. The equation of the linear regression line, along with Pearson’s correlation coefficient (r) or the coefficient of determination (R2), can then be used to evaluate whether your qPCR assay is optimized. This issue may have been addressed in the reference Pfaffl MW. Um novo modelo matemático para quantificação relativa em RT-PCR em tempo 641 real. Pesquisa de ácidos nucleicos. 2001;29(9):e45. Unfortunately, for the authors, the reviewer is much less facile in Portuguese than the authors are in English.
The authors are strongly urged to refer to section 7 in the MIQE guidelines manuscript cited below to address this concern.
2. The concern over the use of a single reference for housekeeping gene.
Other than the issue that -actin is not always stably transcribed under different treatments and growth conditions, there is a concern about the use of a single housekeeping gene. The use of more than one reference or housekeeping gene has been suggested by a recent publication in PLoS ONE. Jacob F, Guertler R, Naim S, Nixdorf S, Fedier A, Hacker NF, et al. (2013) Careful Selection of Reference Genes Is Required for Reliable Performance of RT-qPCR in Human Normal and Cancer Cell Lines. PLoS ONE 8(3): e59180. doi:10.1371/journal.pone.0059180
The authors state “For establishing a set of reference genes for gene normalization we recommend the use of ideally three reference genes selected by at least three stability algorithms.”
The MIQE guidelines (Stephen A. Bustin, Vladimir Benes, Jeremy A. Garson, Jan Hellemans, Jim Huggett, Mikael Kubista, Reinhold Mueller, Tania Nolan, Michael W. Pfaffl, Gregory L. Shipley, Jo Vandesompele, Carl T. Wittwer. The MIQE Guidelines: Minimum Information for Publication of Quantitative Real-Time PCR Experiments, DOI: 10.1373/clinchem.2008.112797 Published March 2009) referenced by Jacob et al., in the 2013 PLoS ONE manuscript included above, state that “Normalization against a single reference gene is not acceptable unless the investigators present clear evidence for the reviewers that confirms its invariant expression under the experimental conditions described. The optimal number and choice of reference genes must be experimentally determined and the method reported.”
Reviewer 3 Report
This manuscript described the use of herbal ATF extract as functional feed additives to reduce oxidative stress in broilers. The manuscript is well written and will be of interest not only the readers of Antioxidants and but also Animals.
I have found a few points that need attention before the manuscript is accepted.
1) Please provide qPCR reaction composition and quantity (line 217).
2) Line 235, as the investigators defined p<0.05 as significance, please use a symbol representing "more than or equal to" for non-significance.
3) I strongly suggest to revise Figure 1 and 2. Perhaps, separate them into several graphs (fig 1a, fig1b, ..) for each gene and each time point. -- The investigators did not compare gene expression between 28d vs 42d. So, I see no point of combining all of them in one difficult-to-read figure.
4) lie 410, remove "surprisingly".
5) Discussion more or less sounds like a literature review to me. In this case, it can be revised to make it more concise. In addition, please consider providing discussion on why the beneficial ATF effects were observed only at 28d, but not at 42d, in serum and liver; why significant changes were found in both 28d and 42d in spleen. I've seen any discussion regarding results in different tested organs and/or serum.
6) line 574; you may want to specify ATF concentration range that gave positive results in this study.
Round 2
Reviewer 2 Report
The extent of the revision and the detail to which the authors addressed the comments is to be commended. The manuscript is ready for publication. Well done.